# Morphology and Morphometry of the Reproductive Tract of the Cotton Boll Weevil after Prolonged Feeding on Alternative Diets

**DOI:** 10.3390/insects14060571

**Published:** 2023-06-20

**Authors:** Thiele da Silva Carvalho, Carlos Alberto Domingues da Silva, Celso Feitosa Martins, Laryssa Lemos da Silva, José Cola Zanuncio, José Eduardo Serrão

**Affiliations:** 1Programa de Pós-Graduação em Zoologia, Universidade Federal da Paraíba, Campus I, Castelo Branco, João Pessoa 58059-900, Brazil; 2Laboratório de Entomologia, Embrapa Algodão, Rua Oswaldo Cruz, 1143, Campina Grande 58428-095, Brazil; 3Departamento de Biologia Geral, Universidade Federal de Viçosa, Viçosa 36570-900, Brazil; 4Departamento de Entomologia, Instituto de Biotecnologia Aplicada à Agropecuária, Universidade Federal de Viçosa, Viçosa 36570-900, Brazil; zanuncio@ufv.br

**Keywords:** *Anthonomus grandis*, atrophy of reproductive tract, *Gossypium hirsutum*, reproductive diapause

## Abstract

**Simple Summary:**

The cotton boll weevil is the main cotton pest in the Americas. Alternative diets induce reproductive dormancy and/or diapause in this weevil. This fact has led researchers to postulate that these boll weevils in reproductive diapause can survive during the off-season, colonizing the subsequent cotton crop. However, there are no data on whether these weevils fed for prolonged periods on alternative food sources can reverse the atrophy of their reproductive organs after being given a diet that favors reproduction, or if their advanced age may impair viable egg production and, consequently, their progeny. The results obtained confirm our hypothesis that feeding for prolonged periods with alternative diets (inappropriate for reproduction) affects the reproductive tract of male and female boll weevils differently, and females and old males may not reverse the atrophy of their reproductive organs even after accessing cotton squares.

**Abstract:**

*Anthonomus grandis* Boheman (Coleoptera: Curculionidae) survives on alternative diets; however, this induces reproductive diapause. The objective was to evaluate the morphology and morphometry of the reproductive tract of this weevil after feeding on alternative diets. The experimental design was completely randomized with 160 replications and treatments arranged in a factorial design 3 × 3, represented by *A. grandis* adults fed on 3 food types (fragments of banana (T1) or orange (T2) endocarp, or with cotton squares of cultivar BRS 286 (T3, control)) and three evaluation periods (30, 60, and 90 days) and after each of these periods they were fed with cotton squares for 10 days. The reproductive tract of 100% of *A. grandis* females fed banana endocarp, orange endocarp, and cotton squares for 30 and 60 days and then cotton squares were morphologically adequate for reproduction, and after 90 days, only 50% of those fed cotton squares were in this condition. The length of the ovarioles and the width of the mature oocyte were greater for *A. grandis* fed on cotton squares and smaller in those with banana and orange endocarps. Histological sections reveal that male testes even with strong degenerative signals are already producing spermatozoa. On the other hand, females displayed ovaries with nurse cells in the tropharium and some maturing oocytes in the vitellarium. The body length was longer and the testis area and diameter smaller in males fed on cotton squares than in those with banana and orange endocarp. *Anthonomus grandis* females fed for ≥90 days with alternative food sources do not recover the functionality of their reproductive tract, even after consuming, for 10 days, a diet that favors reproduction. On the other hand, the males remain with their reproductive organs functional with this condition.

## 1. Introduction

The cotton boll weevil *Anthonomus grandis* Boheman (Coleoptera: Curculionidae) is a key pest of cotton plants in Brazil [1].

The off-season survival of boll weevil adults is well documented in the USA [2,3,4,5] and Brazil [6,7,8]. Boll weevil adults in tropical and subtropical regions survive during the off-season by feeding on remaining cotton plants and/or on alternative diets, such as pollen and parts of native and other cultivated plants [8,9,10,11]. However, these alternative diets induce reproductive dormancy and/or diapause in boll weevils [12,13,14], and they can survive in a state of diapause and/or reproductive dormancy and colonize the subsequent cotton crop. However, there is no data whether these boll weevils fed for prolonged periods with alternative food sources can reverse the atrophy of their reproductive organs after accessing a diet that favors reproduction, or if their advanced age can impair the production of viable eggs and, consequently, their progeny [15]. The duration of the dormancy affects survival and other traits, such as fertility and reproduction, in addition to transgenerational effects reducing the offspring quality [16,17].

The nature of the reproductive dormancy in tropical and subtropical regions has been a matter of debate [9,14,18,19]. The reasoning that dormancy is quiescent [18] is questionable, because the authors did not examine the termination of dormancy [19]. On the other hand, the definition of this reproductive dormancy was based on the observation of fat bodies through the boll weevil abdominal cuticle [14,19]. Furthermore, the diapause was not examined per se [9], and because the insects were field- or trap-collected, neither Guerra et al. [18] nor Greenberg et al. [9] knew the history of the insects examined [19]. The boll weevil dormancy that was defined is a diapause of variable intensity rather than quiescence [19].

Ovaries with condensed oocytes and flocculated yolk, typical of oosorption, and atrophied opaque testes with external fat deposits, filled seminal vesicles, and poorly developed accessory glands were observed in boll weevils undergoing reproductive diapause [13]. The energetic cost of prolonged survival on the morphology and morphometry of the boll weevil’s reproductive organs needs further research. In addition, aging can reduce the production of viable eggs and, consequently, the progeny of this insect [20,21,22].

The hypothesis is that feeding for prolonged periods with alternative diets (unsuitable for reproduction) affects the reproductive tracts of female and male boll weevils differently, with old females and males unable to reverse the atrophy of their reproductive organs even after accessing cotton squares. The objective was to evaluate the morphology and morphometry of the reproductive tract of *A. grandis* females and males fed for prolonged periods with alternative diets and then with cotton squares, simulating a realistic scenario for this insect in the field.

## 2. Materials and Methods

### 2.1. Study Location and Insects

The work was developed at the Laboratory of Entomology of Embrapa Algodão in the municipality of Campina Grande, Paraíba State, Brazil, at 25 ± 2 °C, 60 ± 10% relative humidity and 12 h photophase.

Insects were obtained from cotton squares, cultivar BRS 286, with oviposition punctures and open bracts, in an experimental area of Embrapa Algodão (7°13′ S, 35°54′ W). The collected cotton squares were kept in plastic containers (80 L) until the emergence of *A. grandis* adults. After emergence, 480 adults with the same age were selected, grouped into 10 individuals per 500 mL plastic container covered with a lid, in the proportion of 5 males to 5 females, and kept for 2 hours with water until the beginning of the bioassay. This facilitates handling the insects and optimizes the space available in the climate chamber.

### 2.2. Experimental Design

Boll weevil adults received banana and orange endocarps to evaluate the influence of inadequate food sources in the morphology and morphometry of the reproductive tract of females and males of this insect. These fruits, used as an alternative food by adult boll weevils in southeastern Brazil [7,10,11] in the absence of the main reproductive host (cotton) may replace other alternative food sources inappropriate for reproduction, such as pollen grains from native plants near cotton crops in the Cerrado biome in the Brazilian Midwest region [8,9,10]. In southeastern Brazil, cotton boll weevil adults can feed on orange and banana pulp through cracks, holes, or lesions in the skin of these fruits attached to the canopy or fallen to the ground [7,10,11]. In addition, the cotton boll weevil can feed and survive on orange [10,11] and banana [7] endocarp for long periods of time. The experimental design was completely randomized with 160 replications (boll weevil adults) per treatment in a 3 × 3 factorial design, represented by *A. grandis* adults fed with (T1) banana endocarp fragments (1 cm^3^), (T2) orange endocarp fragments (1 cm^3^), or cotton squares (6 mm in diameter) of cultivar BRS 286 (T3, control) in three evaluation periods (30, 60, and 90 days). Treatments were performed throughout each evaluation period. The parcel consisted of 10 adult boll weevils fed a single food type, which were kept in 100 mL plastic containers with a moistened cotton ball. Food was replaced every two days.

An aliquot of 6 *A. grandis* pairs were separated per treatment and period, with each couple isolated in a 500 mL plastic container, receiving, daily, 3 non-damaged cotton squares of the cultivar BRS 286 (3–6 mm in diameter) for 10 days. These pairs were used for dissection and measurement of male and female reproductive organs. All insects were sexed based on the tergal-notch method of boll weevil sex determination [23]. The non-damaged cotton squares were obtained from cotton plants grown in a greenhouse. The experiment was carried out in duplicate.

### 2.3. Morphological and Morphometric Data

At the end of the bioassay, six pair of boll weevil per treatment and period were dissected to evaluate the morphology and morphometry of their reproductive tracts. The reproductive diapause was identified based on the size and condition of the reproductive tracts of both sexes, with atrophied testes in males and non-activated ovaries in females [12,24].

The reproductive tracts of boll weevils were dissected in 125 mM NaCl, and the ovaries and testes were transferred to Zamboni’s fixative solution [25] and photographed using an Opton digital camera with an image resolution of 13 megapixels, coupled to an El224 stereomicroscope with a 3× objective lens (BEL Equipamentos Analíticos LTDA., Piracicaba, São Paulo, Brazil). The images were transferred to a computer with Image J software Version 1.53s (Rasband, IL, USA; US National Institutes of Health, Bethesda, MD, USA) for morphometric measurements. Adults of this insect from each period evaluated were weighed on an AY220 analytical balance (Shimadzu Corporation, Columbia, MD, USA) with an accuracy of 0.0001 g.

The area and diameter of the testes, the length of the ovarioles, and the width of the most developed oocyte were obtained. The length of the ovarioles and diameter of the testes were used because the size of the insects was similar, reducing variations in their reproductive tracts [24]. Body length was measured from the apex of the rostrum to the end of the elytra, measured along elytral suture [26]. The weight and size of boll weevil males and females were obtained from 25 live pairs. Boll weevil adults were immobilized with surgical forceps and brushes, and their body length measured with aid of a digital caliper INOX 0–150 mm (Fuzhou Winwin Industrial Co., limited, Fujian, China).

The ovaries and testes of the boll weevils were dehydrated in a graded ethanol series (70%, 80%, 90%, and 95%) and embedded in historesin Leica (Leica Biosystems, Heildelberger, Germany). Slices of 3 µm thickness were obtained with glass knives in a rotatory microtome Leica (Leica Biosystems, Heidelberger, Germany) and stained with hematoxylin for 10 min, washed for 10 min in water, stained with aqueous eosin for 30 s, washed in water, and mounted for analysis with light photomicroscope Olympus BX 60 (Olympus, Tokyo, Japan).

### 2.4. Data Analysis

The data of the weight (mg) and the morphometry of body length (mm) of males and females of the boll weevil, length of the ovariole (mm), width of the most developed oocyte (mm) of females, and area and diameter of the male testis were submitted to the Liliefors normality test and transformed into $\sqrt{x+0.5}$. Percentage data were transformed into 1/√(x) [27]. Then, the morphological and morphometric data of the females and males were separately submitted to the two-way analysis of variance (ANOVA), with the means compared using the Tukey test at 5% probability with the Statistical and Genetic Analysis System (SAEG) of the “Universidade Federal de Viçosa”.

## 3. Results

The interaction between the morphological conditions of the reproductive tract and the diet and age of the adult boll weevils was significant (F_4.40_ = 5.00; *p* < 0.001). The morphological features of the reproductive organs of 100% of adult boll weevils fed for 30 and 60 days with cotton squares (Figure 1A and Figure 2A, respectively) or banana (Figure 1C and Figure 2C, respectively) or orange endocarps, and after, each these periods on cotton squares for 10 days were similar. However, after 90 days feeding, the morphological tracts of 100% of males and 50% of females fed on cotton squares (Figure 1B and Figure 2B, respectively) were adequate for reproduction (Table 1 and Table 2), whereas females fed on banana or orange endocarps (Figure 1D and Figure 2D, respectively) showed some atrophy; however, male condition was similar between treatments and ages.

Interactions between the body weight and ovariole length and width of the most developed oocyte after feeding on cotton squares, banana endocarp, or orange endocarp for 30, 60, and 90 days and, after each of these periods, fed on cotton squares for 10 days; the diets previously consumed and the age of the boll weevils were not significant (Table 3). The body weight and length of boll weevil females did not vary with the diet and age, whereas ovariole length and oocyte width varied (Table 3). Oocyte width varied with ovariole length according to the diet, but not with the age of the boll weevil females fed on the same diet.

Ovariole length and oocyte width were greater for boll weevil females fed on cotton squares and smaller with banana or orange endocarps (Table 4). The length of the boll weevil ovarioles at 30 days was greater than that of those at 90 days; however, the oocyte width was similar among treatments.

The interaction between the morphometric parameters of body weight and length and area and diameter of the testes after feeding on cotton squares or banana endocarp or orange endocarp for 30, 60, and 90 days was not significant (Table 3). Male weight did not vary with the diet consumed and age; however, body length and area and diameter of the testis did differ (Table 3). The body length of males fed on cotton squares was smaller, and the testis area and diameter were larger with this diet than with banana or orange endocarps (Table 5).

The histological analyses showed follicles with maturing cysts and some mature spermatozoa in the testes of control males after 30 and 90 days (Figure 3A,B). Males fed on banana or orange endocarps had mature spermatozoa in the testes’ follicles, but spermatids undergoing degeneration in many cysts were revealed by nuclear pyknosis characterized for strong nuclear chromatin condensation, which seems to increase in number from 30 to 90 days (Figure 3C–F).

The histology of females showed ovaries from those in the control and those fed on banana and orange endocarps with a tropharium rich in nurse cells formed a well-developed syncytium with nuclei rich in decondensed chromatin (Figure 4A,D,E). In addition, some maturing oocytes occurred in the vitellarium, with well-developed germinal vesicles (Figure 4B,D,F) delimited by columnar follicular cells (Figure 4B–F), and some patency was evidenced by enlarged intercellular spaces for yolk uptake (Figure 4C).

## 4. Discussion

The interaction between the diet and age of the boll weevils in the morphological conditions of their reproductive organs of the boll weevil indicates that the diet consumed modulates its conditions similar to that reported for this insect fed on alternative diets [28,29,30].

The preserved anatomy of the reproductive organs of male and female boll weevils after 30 and 60 days feeding on alternative diets indicates that this insect may recover from the reproductive diapause after accessing an adequate diet (cotton squares) for reproduction. This was also reported with the termination of the diapause in this insect, although it may remain high after adult feeding on cotton squares [19]. Morphologically preserved reproductive organs in the boll weevils (males and females) fed on cotton squares at 90 days is expected because this food does not induce diapause [31]. In addition, the organs of males fed on banana or orange endocarps for 90 days had smaller testes sizes compared with insects fed on cotton squares, which, when combined with the occurrence of degenerating spermatids, reinforces the hypothesis that unsuitable diets for reproduction during prolonged periods induces the irreversible atrophy of these organs. Therefore, it is likely that reproductive diapause in tropical and subtropical regions only occurs in female boll weevils, which was supported by our histological findings of active tropharium and some developing oocytes in all females, and males with a reduced quantity of mature spermatozoa, although they mate with receptive females, as demonstrated for those of some insect species [32]. Furthermore, the influence of age on the morphological variations in their reproductive organs of the boll weevil, particularly in females, is unknown, but may affect their physiology. The size of the oocytes and the numbers of follicles in the ovarioles and oocytes in the lateral and common oviducts decreased with the age of the rice weevil *Sitophilus oryzae* (Coleoptera: Curculionidae) [15].

Variations in the morphological conditions of the reproductive organs of male and female boll weevils are probably due to the different responses of the first ones fed on alternative diets compared to females and may be related to the risks associated with reproduction in the presence of conflicting host cues [19]. Most female boll weevils (>60%), in temperate climate regions mate in autumn before their wintering diapause. These females retain viable spermatozoa in the spermatheca and lay fertile eggs without additional mating in the spring, although with lower fecundity and fertility [24,33,34,35]. Females in reproductive diapause probably avoid mating with males in diapause, preferring the reproductive ones at the end of this phenomenon to lay viable eggs, ensuring their offspring success. The lower percentage of reproductive active females in an environment without adequate food allocate resources for survival and not for reproduction [19], explaining the lower percentage of reproductively active females. The behavior of males, on the contrary, is to find the host and release a pheromone attracting both sexes [36,37]. Therefore, they may need to remain reproductively active, but this disagrees with the statement that a diet for 7–14 days that favors reproduction in adult boll weevils in reproductive diapause stimulates female reproduction [19]. This is probably due to the differences in the period feeding the boll weevils on alternative diets and then with cotton squares, because those authors fed this insect with a diapause-inducing diet for only 3 to 12 days and then with cotton squares for 14 days [19]. Diet-induced reproductive diapause in adult boll weevils for 3-12 days may not be sufficient to obtain conclusive results, partly explaining the reduced response of this insect to terminate it [19].

The absence of a significant interaction between diet and age for the morphometric parameters of body weight and length, ovariole length, and the width of the most developed oocyte of boll weevil females in reproductive diapause indicates that their fat body reserve is not a parameter, ensuring our diapause findings [19]. This would also explain the lack of variation between the weight and body length of female boll weevils with diet and age. However, the greater ovariole length and oocyte width in females fed on cotton squares compared with those fed with banana or orange endocarp agree with that obtained for the morphology of the boll weevil’s reproductive organs. This indicates that the length of the ovaries [24] is a reliable criterion to determine diapause in boll weevils of a similar size. On the other hand, the greater length of ovarioles in female boll weevils with 30 days feeding with alternative diets compared with 90 days may be related to their aging.

The absence of significant interaction between the diet consumed and age of the males for the morphometric parameters weight and body length and area and diameter of the testes indicates that the accumulation of body fat is not reliable when identifying diapause in male boll weevils [19]. However, the larger area and diameter of the testis in males of this insect fed on cotton squares compared to those with banana or orange endocarps agree with that obtained for the morphology of the boll weevil reproductive organs, but not as a criterion to determining reproductive diapause in its males.

The prolonged period (90 days) with alternative diets unsuitable for reproduction compromised and atrophied the reproductive tract of male and female boll weevils. Furthermore, as female boll weevils aged, their production of viable eggs decreased and, consequently, their success rate for spawning progeny decreased, as evidenced by the morphology and morphometry of their reproductive organs.

## Figures and Tables

**Figure 1 insects-14-00571-f001:**
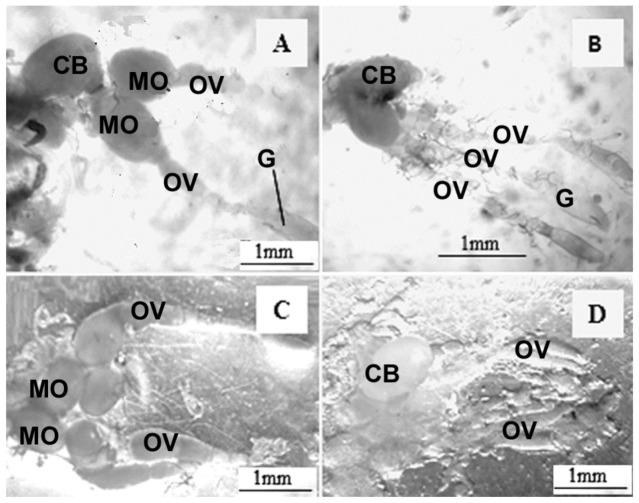
Ovary of female boll weevils fed on cotton squares (**A**) and banana endocarp (**C**) for 30 days and, after each of these periods, with cotton squares for 10 days, showing mature oocytes (MO) in the ovarioles (OV). Fed on cotton squares (**B**) and banana endocarp (**D**) for 90 days and, after each of these periods, with cotton squares for 10 days without absence of mature oocytes in the ovarioles (OV). CB, copulatory bag; G, germarium.

**Figure 2 insects-14-00571-f002:**
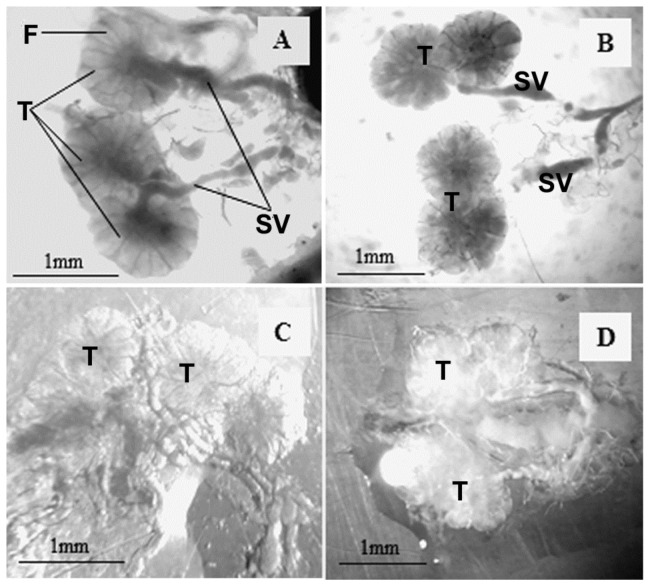
Testis (T) and enlarged seminal vesicles (SV) of male boll weevils fed on cotton squares for 30 (**A**) or 90 days (**B**), on banana endocarp for 30 (**C**) or 90 days (**D**), and after each of these periods with cotton squares for 10 days. F, testis follicle.

**Figure 3 insects-14-00571-f003:**
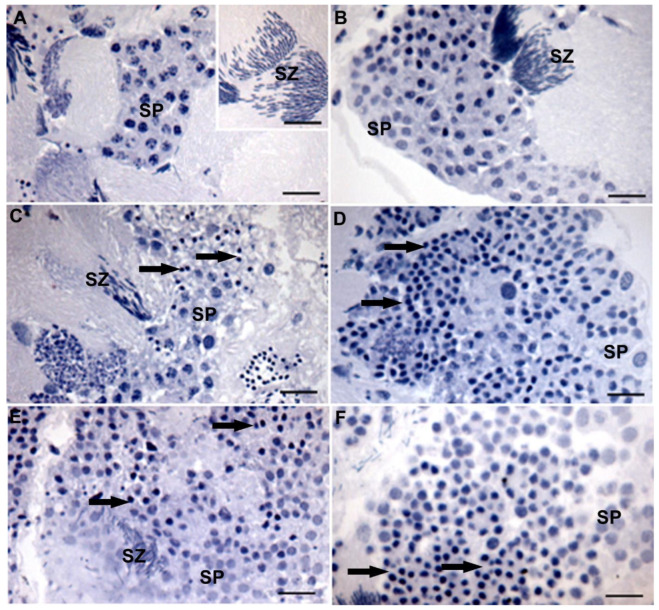
Light micrographs of the testes of *Anthonomus grandis* fed on cotton squares for 30 (**A**) and 90 (**B**) days; banana endocarp for 30 (**C**) and 90 (**D**) days and orange endocarp for 30 (**E**) and 90 (**F**) days showing testis follicles with spermatids (SP) and mature spermatozoa (SZ). Note the increase in degenerating spermatids with a pyknotic nucleus (arrows) from males fed from 30 to 90 days on alternative sources. Scale bars = 10 µm.

**Figure 4 insects-14-00571-f004:**
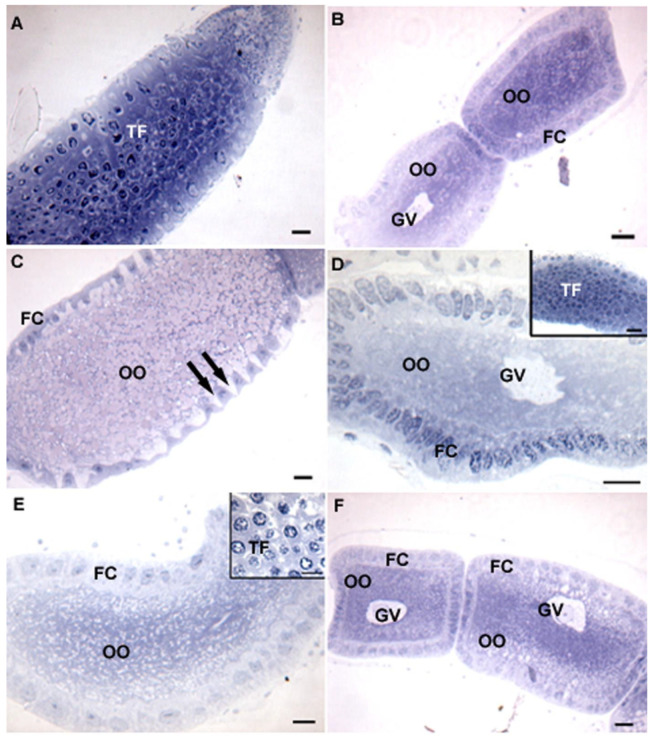
Light micrographs of the ovaries of *Anthonomus grandis* fed on different diets. (**A**) Fed on cotton squares for 30 days showing well-developed tropharium (TF) rich in nurse cells. (**B**) Fed on cotton squares for 90 days showing two developing oocytes (OO) with germinal vesicle (GV) and lined by columnar follicular cells (FC). (**C**) Fed on banana endocarp for 30 days showing developing oocyte (OO) lined by follicular cells (FC) with enlarged intercellular spaces (arrows). (**D**) Fed on banana endocarp for 90 days showing oocyte (OO), germinal vesicle (GV), and follicular cells (FC). Insert: well-developed tropharium (TF). (**E**) Fed on orange endocarp for 30 days with oocyte (OO), follicular cells (FC), and tropharium (TF, insert). (**F**) Fed on orange endocarp for 90 days with two developing oocytes (OO), germinal vesicles (GV), and a follicular epithelium. Scale bars = 10 µm. Insert D = 20 µm.

**Table 1 insects-14-00571-t001:** Summary of the analysis of variance of the percentage of female and male boll weevils with reproductive tract morphologically adequate for reproduction, fed for 30, 60, and 90 days with fragments of banana endocarp, fragments of orange endocarp, and cotton squares and, after each of these periods, fed on cotton squares for 10 days. Campina Grande, Paraíba state, Brazil, 2021.

Sex	Reproductive Tract	Variation Source	DF	MS	*F*	*p*
Female	Morphological conditions	Diet consumed (DC)	2	1666.67	5.00	0.01
		Boll weevil age (BWA)	2	41,666.67	125.00	<0.01
		DC × BWA	4	1666.67	5.00	<0.01
		Residue	40			
Male	Morphological conditions	Diet consumed (DC)	2	-	-	n.s.
		Boll weevil age (BWA)	2	-	-	n.s.
		DC × BWA	4	-	-	n.s.
		Residue	40			

Reproductive tract: data transformed into 1/root(x). DF = Degree of Freedom, MS = Medium Square. Original means showed.

**Table 2 insects-14-00571-t002:** Reproductively active adult weevils (%, mean ± SE) fed for 30, 60, and 90 days on cotton squares, banana endocarp, and orange endocarp and, after each of these periods, on cotton squares for 10 days. Campina Grande, Paraiba State, Brazil, 2021.

Sex	Boll Weevil Age (Days)	Treatments		
		Cotton Squares	Banana Endocarp	Orange Endocarp
Female	30	100.00 ± 0.00 aA	100.00 ± 0.00 aA	100.00 ± 0.00 aA
	60	100.00 ± 0.00 aA	100.00 ± 0.00 aA	100.00 ± 0.00 aA
	90	50.00 ± 22.36 bB	0.00 ± 0.00 bB	0.00 ± 0.00 bB
Male	30	100.00 ± 0.00 aA	100.00 ± 0.00 aA	100.00 ± 0.00 aA
	60	100.00 ± 0.00 aA	100.00 ± 0.00 aA	100.00 ± 0.00 aA
	90	100.00 ± 0.00 aA	100.00 ± 0.00 aA	100.00 ± 0.00 aA

Means followed by the same small letter per row or capital letter per column do not differ by Tukey test (*p* ≤ 0.05).

**Table 3 insects-14-00571-t003:** Summary of the analysis of variance for weight (mg) and body length (mm) of female and male boll weevils; ovariole length (mm) and width of the most developed oocyte (mm) of females, and area and diameter of male testis, fed for 30, 60, and 90 days with fragments of banana endocarp, fragments of orange endocarp, and cotton squares and, after each of these periods, on cotton squares for 10 days. Campina Grande, Paraíba, Brazil, 2021.

Sex	Morphometric Data	Variation Source	DF	MS	*F*	*p*
Female	Weight	Diet consumed (DC)	2	1.41 × 10^−5^	0.46	>0.05
		Boll weevil age (BWA)	2	1.19 × 10^−5^	0.39	>0.05
		DC × BWA	4	3.19 × 10^−5^	1.04	=0.40
		Residue	45	3.07 × 10^−5^		
	Body length	Diet consumed (DC)	2	0.45 × 10^−1^	0.43	>0.05
		Boll weevil age (BWA)	2	0.92 × 10^−1^	0.87	>0.05
		DC × BWA	4	0.70 × 10^−1^	0.66	>0.05
		Residue	45	0.11		
	Ovariole length	Diet consumed (DC)	2	4.44	25.11	<0.01
		Boll weevil age (BWA)	2	0.99	5.61	<0.01
		DC × BWA	4	0.19	1.10	=0.37
		Residue	45	0.18		
	Width of most developed oocyte	Diet consumed (DC)	2	0.39 × 10^−1^	12.67	<0.01
		Boll weevil age (BWA)	2	0.23 × 10^−4^	0.01	>0.05
		DC × BWA	4	0.15 × 10^−2^	0.49	>0.05
		Residue	45	0.31 × 10^−2^		
Male	Weight	Diet consumed (DC)	2	0.59 × 10^−5^	2.71	=0.08
		Boll weevil age (BWA)	2	0.19 × 10^−5^	0.86	>0.05
		DC × BWA	4	0.51 × 10^−5^	2.35	=0.07
		Residue	45	0.22 × 10^−5^		
	Body length	Diet consumed (DC)	2	0.48	7.09	<0.01
		Boll weevil age (BWA)	2	0.15	2.19	=0.12
		DC × BWA	4	0.13	1.93	=0.12
		Residue	45	0.67		
	Testis area	Diet consumed (DC)	2	0.47	49.34	<0.01
		Boll weevil age (BWA)	2	0.28	29.68	<0.01
		DC × BWA	4	0.42	0.45	>0.05
		Residue	45	0.95		
	Testis diameter	Diet consumed (DC)	2	0.30	39.83	<0.01
		Boll weevil age (BWA)	2	0.18	29.94	<0.01
		DC × BWA	4	0.42	0.56	>0.05
		Residue	45	0.75		

DF = Degree of Freedom, MS = Mean Square.

**Table 4 insects-14-00571-t004:** Weight (mg) and body length (mm), ovariole length (mm), and width of the most developed oocyte (mm) (mean * ± standard error) of boll weevil females fed for 30, 60, and 90 days with fragments of banana endocarp, fragments of orange endocarp, and cotton squares and, after each of these periods, with cotton squares for 10 days. Campina Grande, Paraíba, Brazil, 2021.

Morphometric Data	Treatments		
	Banana Endocarp	Orange Endocarp	Cotton Squares
Weight	16.00 ± 0.40 a	16.22 ± 0.39 a	15.67 ± 0.44 a *
Body length	4.08 ± 0.08 a	4.11 ± 0.09 a	4.01 ± 0.05 a
Ovariole length	1.96 ± 0.11 b	2.02 ± 0.11 b	2.85 ± 0.11 a
Width of most developed oocyte	0.30 ± 0.01 b	0.30 ± 0.01 b	0.38 ± 0.02 a
	Boll weevil age (days)		
	30	60	90
Weight	16.11 ± 0.41 a	16.11 ± 0.38 a	15.67 ± 0.44 a
Body length	4.15 ± 0.08 a	4.05 ± 0.07 a	4.01 ± 0.08 a
Ovariole length	2.51 ± 0.12 a	2.28 ± 0.12 ab	2.04 ± 0.16 b
Width of most developed oocyte	0.32 ± 0.02 a	0.32 ± 0.02 a	0.32 ± 0.01 a

(*) Means followed by the same letter in the line do not differ by Tukey test (*p* ≤ 0.05).

**Table 5 insects-14-00571-t005:** Weight (mg) and body length (mm), and area (mm^2^) and testis diameter (mm) (means * ± standard error) of male boll weevils fed for 30, 60, and 90 days with fragments of banana endocarp, fragments of orange endocarp, and cotton squares and, after each of these periods, with cotton squares for 10 days. Campina Grande, Paraíba State, Brazil, 2021.

Morphometric Data	Treatments		
	Banana Endocarp	Orange Endocarp	Cotton Squares
Weight	15.80 ± 0.40 a	15.56 ± 0.35 a	14.78 ± 0.33 a *
Body length	4.14 ± 0.08 a	4.07 ± 0.06 a	3.85 ± 0.05 b
Testis area	0.56 ± 0.04 b	0.56 ± 0.03 b	0.85 ± 0.03 a
Testis diameter	0.91 ± 0.03 b	0.87 ± 0.03 b	1.12 ± 0.02 a
	Boll weevil age (days)		
	30	60	90
Weight	15.78 ± 0.42 a	15.22 ± 0.37 a	15.20 ± 0.32 a
Body length	4.13 ± 0.08 a	3.98 ± 0.06 a	3.97 ± 0.07 a
Testis area	0.78 ± 0.04 a	0.68 ± 0.04 b	0.52 ± 0.04 c
Testis diameter	1.08 ± 0.03 a	0.96 ± 0.03 b	0.87 ± 0.03 c

(*) Means followed by the same letter in the line do not differ by Tukey test (*p* ≤ 0.05).

## Data Availability

Data are contained within the article.

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
