# Peer review of "Morphology and Morphometry of the Reproductive Tract of the Cotton Boll Weevil after Prolonged Feeding on Alternative Diets"

_insects, 2023, doi:10.3390/insects14060571_

Round 1

Reviewer 1 Report

1)      In line 104, the authors “assume” and state that bananas and oranges are not commonly used as food by the boll weevil…this statement depreciates the relevance of the study. The study would be much more relevant and important if they used alternative diets that the boll weevil is known to feed on in Brazil.

2)      Given that the authors waited for adults to emerge from squares…the actual age of adult weevils used in the experiments is not known (some weevils may remain in squares for up to several days before emerging). Further, the feeding regime during the first three days following adult eclosion is critical and greatly influences the reproductive/diapause status of weevils…it appears weevils were provided only water during the first three days? How often were food items replaced during the 30-, 60-, and 90-day feeding periods? These details need to be provided. Based on my experience, you also start to see high levels of mortality if weevils are not provided any food. What percentage of weevils died for each feeding treatment (diet by duration combination)? In my experience, you begin to see substantial mortality after 10 or so days even for weevils provided an optimal diet (e.g.,squares) that go reproductive. I suspect that for the weevils to survive the 30-, 60-, and 90-day feeding period, most had become reproductively dormant and accumulated fat. Ideally, the authors should have dissected a subset of weevils from each diet by duration combination to assess reproductive status so they would have an idea what proportion of weevils were reproductively dormant or active prior to placing them on the square diet for 10 days.

3)      Line 110: The authors indicate there were 160 replications but it is unclear what constitutes an experimental replication in the study. In line 97, the authors indicate weevils were held in groups of 10 weevils.  What was the purpose of this? Why not assign weevils individually to treatments as they became available. Were 480 weevils available at one time, or were weevils grouped in 10s as they emerged from squares? That is, were treatments assigned over time? One group of 10 assigned to one treatment, and then next group of 10 weevils assigned to another treatment? This should be clarified. Based on lines 115-120, even though the experiment was duplicated it appears the entire sample size for the study was only 24 weevils (12 males and 12 females) for each treatment combination (food type by evaluation period). This sample size is too small to make any valid inferences, particularly when looking at percentages/proportions. If the authors started with 480 weevils in each run of the experiment and split them evenly among the six feeding treatments, that would yield 80 (40 males and 40 females) weevils per treatment. Why were only six males and six females dissected from each treatment? Why not dissect and assess reproductive status for all the weevils from each treatment to increase sample size? Was there considerable weevil mortality during the feeding evaluation periods so only six couples were available for dissection? With a larger sample size, the authors would be able to conduct probability statistics to determine the probability of weevils becoming reproductive following the different feeding treatments (diet x duration) and after feeding on squares for 10 days. Again, unless I misinterpreted the experimental design/set up, the sample size is too small to make any valid conclusions.

4)      Line 132: In my experience, body weight can be misleading because it is greatly influenced by recency of feeding and/or excretion…measurement of body size is better.

5)      Tables 1 and 3: Unless I misinterpreted the experimental set up (sample size), the residuals do not match up. I suspect there was considerable mortality within each feeding duration period.

6)      Figure 1: clarity is unacceptable for publication…need better pictures.

7)      Figure 2 is better, but clarity still needs to be improved for publication.

8)      I don’t provide comments on the Results and Discussion sections because there are too many unanswered questions regarding the experimental design so it’s difficult to determine if their interpretation of results is valid.

Author Response

RESPONSE TO THE REVIEWER 01

1) Reviewer comments: In line 104, the authors “assume” and state that bananas and oranges are not commonly used as food by the boll weevil…this statement depreciates the relevance of the study. The study would be much more relevant and important if they used alternative diets that the boll weevil is known to feed on in Brazil.

Authors' response: We agree with the reviewer and for this reason, we have reformulated the text to improve the reader's understanding. It is a fact, the use  of alternative diets that cotton boll weevil usually consume in the Brazilian Cerrado is important. However, in the southeastern region of Brazil, where cotton is also cultivated, the presence of adjacent areas cultivated with banana and orange is expressive. In these places, the adult cotton boll weevil can feed on orange and banana pulp through cracks, holes or lesions in the skin of these fruits attached to the crown or fallen to the ground. In the Brazilian Cerrado this does not occur, which is why we state that weevils do not usually feed on the fruits of these plants, since cotton cultivation in the Brazilian Cerrado represents 90% of the area planted with this mallow in the country. On the other hand, it is important to point out that, with the exception of cotton squares and smalls bolls, no other alternative diet (pollen from other plants) commonly used by the cotton boll weevil in the Brazilian Cerrado constitutes adequate food for its survival and reproduction. Therefore, respectfully, we believe that this does not compromise the study, as both the pollen of alternative plants found in the forests surrounding the areas cultivated with cotton, as well as banana and orange fruits, are inadequate food sources used by this pest.

2)

2.1) Reviewer comments: Given that the authors waited for adults to emerge from squares…the actual age of adult weevils used in the experiments is not known, (some weevils may remain in squares for up to several days before emerging).

Authors' response: A caveat is in order here; the boll weevils have been fed since their emergence from cotton squares with at least one of the three diets informed in the study. We consider that boll weevils remain for a few days (2-3 days) inside cotton square before emerging. For this reason, we inform you that these are 3-day-old boll weevils.

2.2) Reviewer comments: Further, the feeding regime during the first three days following adult eclosion is critical and greatly influences the reproductive/diapause status of weevils…it appears weevils were provided only water during the first three days?

Authors' response: We appreciate the reviewer's remarks on the age of the weevils. Therefore, we reformulated the text to make it clearer for the reader. However, we emphasize that the boll weevils did not receive water only in the first three days. They were fed, as mentioned, since leaving the cotton squares, at least one of the three diets reported in the study.

2.3) Reviewer comments: How often were food items replaced during the 30-, 60-, and 90-day feeding periods?

Authors' response: Food was replaced every two days.

2.4) Reviewer comments: These details need to be provided.

Authors' response: We have modified this part of the manuscript text to provide the details requested by the reviewer.

2.5) Reviewer comments: Based on my experience, you also start to see high levels of mortality if weevils are not provided any food. What percentage of weevils died for each feeding treatment (diet by duration combination)? In my experience, you begin to see substantial mortality after 10 or so days even for weevils provided an optimal diet (e.g.,squares) that go reproductive. I suspect that for the weevils to survive the 30-, 60-, and 90-day feeding period, most had become reproductively dormant and accumulated fat. Ideally, the authors should have dissected a subset of weevils from each diet by duration combination to assess reproductive status so they would have an idea what proportion of weevils were reproductively dormant or active prior to placing them on the square diet for 10 days.

Authors' response: We agree with the reviewer that, ideally, it would be interesting to dissect a subset of boll weevils from each diet by duration combination to assess reproductive status, before placing them on the cotton squares diet for 10 days. However, this study was part of a PhD thesis where, in addition to the data contained in this article, other data on the survival and fecundity of the insect were determined. These data were used in the preparation of another manuscript submitted for publication in another indexed journal. Therefore, we did not have the conditions and/or sufficient number of boll weevils to meet the ideally proposed by the reviewer, because we would have to sacrifice more individuals for this purpose, compromising the other results proposed in the study. However, we believe that the dissection of boll weevils after this insect consumes cotton squares for 10 days, also, perfectly meets the purposes of this study.

3)

3.1) Reviewer comments: Line 110: The authors indicate there were 160 replications but it is unclear what constitutes an experimental replication in the study. In line 97, the authors indicate weevils were held in groups of 10 weevils. What was the purpose of this?

Authors' response: Thanks for the reviewer's questioning about our purpose in grouping 10 boll weevils and inform that this was done simply to facilitate the handling of insects and optimize the space available in the climate chamber.

3.2) Reviewer comments: Why not assign weevils individually to treatments as they became available?

Authors' response: We appreciate the reviewer's suggestion, but we feel that it is unnecessary to assign boll weevils individually to treatments as they become available because all boll weevils that made up the treatment were available at the same time.

3.3) Reviewer comments: Were 480 weevils available at one time, or were weevils grouped in 10s as they emerged from squares?

Authors' response: Yes, the 480 boll weevils were available at the same time.

3.4) Reviewer comments: That is, were treatments assigned over time?

Authors' response: Yes. Treatments with boll weevil adults fed on cotton squares, banana pulp and/or orange pulp were assigned over time.

3.5) Reviewer comments: One group of 10 assigned to one treatment, and then next group of 10 weevils assigned to another treatment?

Authors' response: No, because all adult boll weevils after undergoing treatment with each specific diet by assessment date were the same age.

3.6) Reviewer comments: This should be clarified.

Authors' response: We modified this part of the text of the manuscript to clarify the doubts raised by the reviewer.

3.6) Reviewer comments:  Based on lines 115-120, even though the experiment was duplicated it appears the entire sample size for the study was only 24 weevils (12 males and 12 females) for each treatment combination (food type by evaluation period). This sample size is too small to make any valid inferences, particularly when looking at percentages/proportions. If the authors started with 480 weevils in each run of the experiment and split them evenly among the six feeding treatments, that would yield 80 (40 males and 40 females) weevils per treatment. Why were only six males and six females dissected from each treatment? Why not dissect and assess reproductive status for all the weevils from each treatment to increase sample size? Was there considerable weevil mortality during the feeding evaluation periods so only six couples were available for dissection? With a larger sample size, the authors would be able to conduct probability statistics to determine the probability of weevils becoming reproductive following the different feeding treatments (diet x duration) and after feeding on squares for 10 days. Again, unless I misinterpreted the experimental design/set up, the sample size is too small to make any valid conclusions.

Authors' response: We agree with the reviewer that using twelve couples maybe a small sample for l morphological measurements. However, we have omitted from the text of the methodology, by inattention, that  weight and body size of boll weevils were obtained using 25 couples, as for this type of measurement it is not necessary to sacrifice and/or kill the insects. In fact, the 12 sacrificed couples were used only for dissecting the insects and measuring their reproductive organs. Therefore, we consider our sample size valid. In addition, the feeding treatments were nine and not six (the reviewer forgot about the control, boll weevil fed with cotton squares). Therefore, if we consider that the sample size for each treatment is 160 boll weevils equally divided into treatments for each evaluation date, we would have 53 individuals, so 24 sampled individuals represent approximately 45% of the population of boll weevils analyzed. This is a sample of 160 individuals with homogeneous data and statistically reliable, as we are analyzing 4 variables (length of the ovariole and width of the most developed oocyte for females and testicular area and diameter for males) in a factorial scheme, the minimum recommended would be 20 subjects (at least 5 times more observations than the number of variables). As we used 24 individuals, this requirement was satisfied for the analyzed dataset (Matos and Rodrigues, 2019). In addition, the experiment was conducted in duplicate, which indicates that the results obtained in a single replication were confirmed in the second replication. Therefore, in our humble opinion, this does not invalidate our results.

MATOS, D.A.S.; RODRIGUES, E.C. Factor analysis. Brasília: ENAP. 2019. Available in: http://repositorio.enap. gov.br/handle/1/4790

4) Reviewer comments (Line 132): In my experience, body weight can be misleading because it is greatly influenced by recency of feeding and/or excretion…measurement of body size is better.

Authors' response: We agree with the reviewer, but it is important to note that in addition to the body weight of the boll weevils, we also evaluated body size.

5) Reviewer comments: Tables 1 and 3: Unless I misinterpreted the experimental set up (sample size), the residuals do not match up. I suspect there was considerable mortality within each feeding duration period.

Authors' response: We agree with the reviewer that a considerable mortality within each feeding period occurred, but the residual values are correct! In the case of ANOVA in factorial designs. The sums of squares remain the same, but the residual sum of squares and the associated degrees of freedom increase, as the sum of squares of the removed factors are incorporated into the residuals. Reminding the reviewer that this study did not aim to evaluate boll weevil survival.

6) Reviewer comments: Figure 1: clarity is unacceptable for publication…need better pictures.

Authors' response: We appreciate the reviewer's suggestion and try to improve the quality of the figures as much as possible. In the case of figure 1, if the reviewer does not agree that the changes we implemented in figure 1 are sufficient, we can add it to the work, as supplementary material. However, we are sure the reviewer will approve figure 2, as the quality has greatly improved after tweaks.

7) Reviewer comments: Figure 2 is better, but clarity still needs to be improved for publication.

Authors' response: our comment for this item is the same as for item 6.

8) Reviewer comments: I don’t provide comments on the Results and Discussion sections because there are too many unanswered questions regarding the experimental design so it’s difficult to determine if their interpretation of results is valid.

Authors' response: the comment was not answered in this item because there were no questions from the reviewer but we revised and improved these sections of the manuscript.

Reviewer 2 Report

The authors studied the influence of an inappropriate diet on the reproductive ability based on morphological changes of reproductive organs in both sexes of Anthonomus grandis Boheman, 1843.

Generally: It should be clearly explained that there is a basic difference between a nutritional feeding (imagoes), here purely artificial having nothing to do with the real nutritional feeding of this species in the nature, and the real host plant of this species (larvae) in its natural habitat. The MS completely lacks comments on the above two important types of feeding. I recommend to complete.

Although I am not a native English speaker, I urgently recommend to check the text as English grammar, punctuation and stylistics concerns.

Comments/Suggestions:

Lines 66–73: I suggest to conclude the definition of the term “reproductive dormancy” as currently understood in order to work with well defined physiological terms.

Lines 86–87: The simulation of the “realistic scenario” in natura is here purely hypothetical, as the diet in this study concerns exclusively the nutritional feeding, not the true host plant of this sp. of Curculionoidea. This fact should be pointed out at least in this part of the text.

Lines 138–139: The body length is being always measured in Curculionoidea from the rostrum base to the apex of elytra, i.e., rostrum excluded. The reason is simple: rostrum is the most variable part of a body in most species of Curculionoidea, especially of those belonging to Curculionidae (Curculioninae) comprising the tribe Anthonomini.

I recommend this ms for the publication provided the comments will be implemented and English will be checked.

Author Response

RESPONSE TO THE REVIEWER 02

The authors studied the influence of an inappropriate diet on the reproductive ability based on morphological changes of reproductive organs in both sexes of Anthonomus grandis Boheman, 1843.

1) Reviewer comments: Generally: It should be clearly explained that there is a basic difference between a nutritional feeding (imagoes), here purely artificial having nothing to do with the real nutritional feeding of this species in the nature, and the real host plant of this species (larvae) in its natural habitat. The MS completely lacks comments on the above two important types of feeding. I recommend to complete.

Authors' response: We respectfully disagree with the reviewer, as the adult cotton boll weevil can feed on orange and banana pulp through cracks, holes or lesions in the skin of these fruits attached to the canopy or fallen on the ground in areas near cotton plantations. The study would have been more complete if we had used alternative diets that weevils usually consume in the Brazilian Cerrado. However, in the southeastern region of Brazil, where cotton is also cultivated, the presence of adjacent areas cultivated with bananas and oranges is expressive. In the Brazilian Cerrado this does not occur, which is why we state that weevils do not usually feed on the fruits of these plants, since cotton cultivation in the Brazilian Cerrado represents 90% of the area planted with this mallow in the country. On the other hand, it is important to emphasize that, with the exception of squares and small cotton capsules, no other alternative diet, pollen from other plants or pulp from banana and orange fruits, used by the cotton boll weevil are suitable foods for their survival and reproduction. Therefore, the treatments in which boll weevil adults feed on the pulp of banana and orange fruits are not artificial and actually occur in practice under field conditions.

2) Reviewer comments: Although I am not a native English speaker, I urgently recommend to check the text as English grammar, punctuation and stylistics concerns.

Authors' response: We appreciate the reviewer's suggestion and we reviewed the English of this manuscript.

3) Reviewer comments (Lines 66–73): I suggest to conclude the definition of the term “reproductive dormancy” as currently understood in order to work with well defined physiological terms.

Authors' response: We believe that it is unnecessary to define the term “reproductive dormancy”, as this article does not intend to debate and define whether the boll weevil enters “reproductive dormancy” or “reproductive diapause”. We just want to show that the boll weevil, when surviving the off-season feeding on alternative food sources, can infest the subsequent cotton crop, but only for a certain period.

4) Reviewer comments (Lines 86–87): The simulation of the “realistic scenario” in natura is here purely hypothetical, as the diet in this study concerns exclusively the nutritional feeding, not the true host plant of this sp. of Curculionoidea. This fact should be pointed out at least in this part of the text.

Authors' response: Yes, the study did not intend to evaluate the reproductive performance of the boll weevil in its natural host (cotton squares), as this is already well known.

5) Reviewer comments (Lines 138–139): The body length is being always measured in Curculionoidea from the rostrum base to the apex of elytra, i.e., rostrum excluded. The reason is simple: rostrum is the most variable part of a body in most species of Curculionoidea, especially of those belonging to Curculionidae (Curculioninae) comprising the tribe Anthonomini.

Authors' response: We agree with the reviewer that correct measurement of body length in Curculionoidea should not exclude the insect's rostrum. These parameters were adopted to measure the size of the boll weevil.

Reviewer 3 Report

The authors turned to the very important problem – crop defense from pests. The work is done on the fundamental level – the authors have studied the possibility of pest expansion into the new habitats, then – the possibility of cotton fields damage. They used original and at the same time understandable methods to estimate feeding proposals in cotton boll weevil and its reproduction system. Moreover, the work contributes greatly into fundamental problems of sexual dimorphism variation and its manifestation in different taxa and different traits.

I have no any major notes.

Minor notes are as follows:

1.      Why did the authors use the term “tract” in the title of MS? I think that “trait” will be more adequate.

2.      References. N 32 lack the year of publication.

Author Response

RESPONSE TO THE REVIEWER 03

The authors turned to the very important problem – crop defense from pests. The work is done on the fundamental level – the authors have studied the possibility of pest expansion into the new habitats, then – the possibility of cotton fields damage. They used original and at the same time understandable methods to estimate feeding proposals in cotton boll weevil and its reproduction system. Moreover, the work contributes greatly into fundamental problems of sexual dimorphism variation and its manifestation in different taxa and different traits.

I have no any major notes.

Minor notes are as follows:

1) Reviewer comments: Why did the authors use the term “tract” in the title of MS? I think that “trait” will be more adequate.

Authors' response: Thanks for the reviewer's suggestion. However, we believe the term “tract” used in the article is correct.

2) Reviewer comments: References. N 32 lack the year of publication.

Authors' response: We thank the reviewer for pointing out the error in bibliographic reference number 32 and we inform you that we have made the necessary corrections as requested.

Round 2

Reviewer 1 Report

For the most part, the authors adequately addressed my comments but I'm not sure the additional text adequately addresses them. Also, if all weevils were available at one time, why was it necessary to assign treatments over time?

Author Response

RESPONSE TO THE REVIEWER 01

1) Reviewer comments: For the most part, the authors adequately addressed my comments but I'm not sure the additional text adequately addresses them. Also, if all weevils were available at one time, why was it necessary to assign treatments over time?

Authors' response: We revised and improved the text of the manuscript to respond as best as possible to the reviewer's comments, including adding information in red. It is true that all boll weevils were available at the same time, but for each period, an aliquot of boll weevils (30, 60 and 90 days) was examined. Therefore, whether body morphometric and morphological measurements of adult boll weevils and their reproductive systems (males and females) were performed at 30, 60 and 90 days, this, in our opinion, can only be determined over time.
